# Phase Transformation and Characterization of 3D Reactive Microstructures in Nanoscale Al/Ni Multilayers

**Yesenia Haydee Sauni Camposano** [1,*], **Sascha Sebastian Riegler** [2], **Konrad Jaekel** [3], **Jörg Schmauch** [4], **Christoph Pauly** [5], **Christian Schäfer** [5], **Heike Bartsch** [3], **Frank Mücklich** [5], **Isabella Gallino** [2] **and Peter Schaaf** [1]

1   Materials for Electrical Engineering and Electronics Department, Institute of Materials Science and Engineering, Institute of Micro and Nanotechnology MacroNano®, TU Ilmenau, Gustav-Kirchhoff-Str. 5, 98693 Ilmenau, Germany; peter.schaaf@tu-ilmenau.de

2   Metallic Materials Department, Saarland University, Campus C6.3, 66123 Saarbrücken, Germany; sascha.riegler@uni-saarland.de (S.S.R.); i.gallino@mx.uni-saarland.de (I.G.)

3   Electronics Technology Group, Institute of Materials Science and Engineering, Institute of Micro and Nanotechnology MacroNano®, TU Ilmenau, Gustav-Kirchhoff-Str. 1, 98693 Ilmenau, Germany; konrad.jaekel@tu-ilmenau.de (K.J.); heike.bartsch@tu-ilmenau.de (H.B.)

4   Physics Department, Saarland University, Campus D2.2, 66123 Saarbrücken, Germany; schmauch@nano.uni-saarland.de

5   Functional Materials Department, Saarland University, Campus D3.3, 66123 Saarbrücken, Germany; c.pauly@mx.uni-saarland.de (C.P.); christian.schaefer@uni-saarland.de (C.S.); muecke@matsci.uni-sb.de (F.M.)

*   Correspondence: yesenia.sauni@tu-ilmenau.de; Tel.: +49-3677-69-3274

**Abstract:** Reactive multilayer systems represent an innovative approach for potential usage in chip joining applications. As there are several factors governing the energy release rate and the stored chemical energy, the impact of the morphology and the microstructure on the reaction behavior is of great interest. In the current work, 3D reactive microstructures with nanoscale Al/Ni multilayers were produced by alternating deposition of pure Ni and Al films onto nanostructured Si substrates by magnetron sputtering. In order to elucidate the influence of this 3D morphology on the phase transformation process, the microstructure and the morphology of this system were characterized and compared with a flat reactive multilayer system on a flat Si wafer. The characterization of both systems was carried out before and after a rapid thermal annealing treatment by using scanning and transmission electron microscopy of the cross sections, selected area diffraction analysis, and differential scanning calorimetry. The bent shape of multilayers caused by the complex topography of silicon needles of the nanostructured substrate was found to favor the atomic diffusion at the early stage of phase transformation and the formation of two intermetallic phases $Al_{0.42}Ni_{0.58}$ and $AlNi_3$, unlike the flat multilayers that formed a single phase AlNi after reaction.

**Keywords:** reactive multilayers; black silicon; self-propagating reactions; phase transformation; sputtering; aluminum/nickel; rapid thermal annealing

## 1. Introduction

Reactive multilayer systems (RMS), typically forming binary or ternary intermetallic phases, have been thoroughly investigated due to their energetic properties and potential applications [1]. RMS consist of alternating layers of two or more components which react exothermically when a localized pulse of energy is applied. The released heat of the reaction promotes the reaction in the neighboring zones, establishing a self-sustained and self-propagating reaction [1,2]. Due to the high amount of stored chemical energy and the large energy release rate of RMS, they can be used in technological applications such as welding, brazing, or in thermal batteries [3–6]. More recently, Al/Ni RMS were used as an ultrafast heat source to produce high entropy alloy films [7]. Therefore, much

effort has been made in order to study the influence of different RMS characteristics, such as bilayer thickness, intermixing thickness, and chemical composition, on its energetic properties [4,8–11]. For the equiatomic Al/Ni RMS, the phase transformation process has already been widely investigated in literature [12]. During a self-propagating reaction, the formation of AlNi intermetallic phase occurs directly from a semi liquid–solid state [13]. However, during annealing with low heating rates the formation of $Al_9Ni_2$ or $Al_3Ni$ at the early stage is observed, and later a sequence of phases with the formation of AlNi as the final product [12,14]. Recently, more attention has been given to the substrate topography, since it will impact the interface roughness of the RMS and on the phase transformation [15]. A rough interface with a high density of defects favors the atomic diffusion of Ni in Al [15,16]. These studies were carried out using multilayer foils. However, it is also possible to produce reactive multilayer particles by using structured substrates such as nylon fibers or black Si [17,18]. Particles produced on nylon fibers were separated from the substrate and funneled into glass tubes in order to measure the propagation front velocity. It was 200 times slower than a RMS-foil with the same characteristics and the slower propagation was attributed to a reduction of the heat transfer due to the thermal resistances at the particle–particle interfaces [17,19]. Nanostructured Si has interesting properties such as surface enlargement that can be exploited in chip assembly [20]. For this reason, it was used as a substrate during the deposition of Al/Ni RMS [18]. The needle-shaped topography of Si disturbs the morphology of the deposited multilayers, giving rise to a complex structure of 3D reactive multilayers (3D-RMS). Furthermore, the black Si surface enhanced the adhesion of the RMS on the substrate in comparison with a flat surface, even after heat treatment [18]. During rapid thermal annealing (RTA) at a maximum temperature of 550 °C, 3D-RMS on Si needles reacted only partially forming B2-AlNi leaving unreacted Ni residue, unlike RMS foils on flat Si which fully transform the precursor materials into the B2-AlNi phase. The complex morphology of microscale structures with nanoscale reactive layers and its influence in the phase transformation and energetic properties were not studied thus far in literature to our knowledge; hence, this paper is focused on the study of the novel morphology of multilayers caused by the silicon needles and its influence on the phase transformation process. By using scanning and transmission electron microscopy (SEM/TEM), geometric parameters of the microstructures were calculated. Local morphology before and after RTA is investigated by X-ray diffraction analysis (XRD) and selected area electron diffraction (SAED). All results are compared with results from differential scanning calorimetry (DSC) experiments.

## 2. Materials and Methods

### 2.1. Sample Preparation

　　In order to obtain RMS with a 3D morphology, a silicon wafer with a nanostructured surface (black Si) was used as a substrate. Silicon needles were prepared using p-type doped <004> silicon wafers that were processed by reactive ion etching (RIE). This process was carried out by using a RIE system PlasmaLab 100 Oxford, where the working pressure and the plasma power were set as 13.3 Pa and 100 W, respectively. During the process helium backing was used, the temperature was kept at 20 °C, and the flow rates of $SF_6$ and $O_2$ were 84 sccm and 66 sccm, respectively. The etching time was 30 min resulting in wafers with needles approximately 1.0 μm in length on the Si surface. The morphology of the nanostructured Si obtained is depicted in Figure 1.

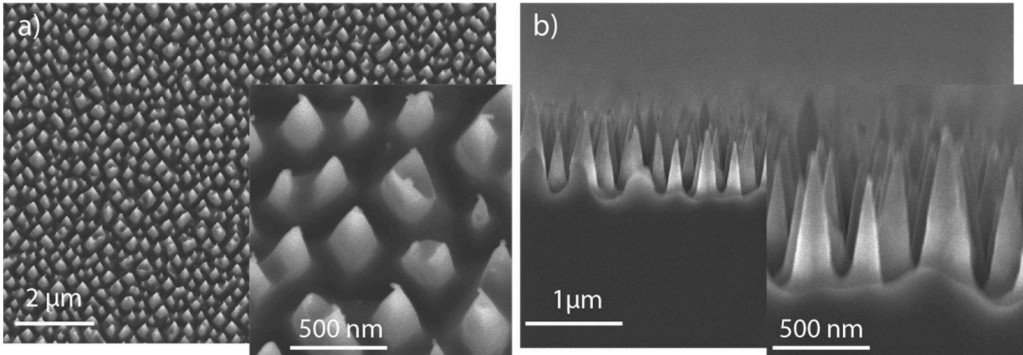

**Figure 1.** Scanning electron microscopy images of the nanostructured Si substrate: (**a**) top view of the silicon needles with a tilt angle of 15°; (**b**) cross section of the substrate providing a side view of the silicon needles.

For the purpose of comparing the resulting 3D-RMS structures, a flat p-type doped <111> Si wafer substrate was used as reference. In this study, the difference in Si wafer orientations does not represent a critical factor. Al/Ni RMS deposition by direct current (DC) magnetron sputtering was performed directly on both types of substrates using the cluster deposition system (CS400 by von Ardenne), with the substrate-to-target distance of 105 mm, a working pressure of 0.5 Pa, a sputtering power of 200 W, and an argon flow of 80 sccm at room temperature, resulting in a deposition rate of 0.32 nm/s for aluminum and 0.26 nm/s for nickel. Two targets, both with a diameter of 100 mm, were used: Al (99.99% purity, FHR) and Ni (99.99% purity, FHR). DC sputtering deposition was carried out until reaching 100 Al/Ni bilayers with layers thicknesses of 20 nm Al and 20 nm Ni. (The individual layer thickness was achieved by fixed deposition time per layer under consideration of the previously determined growth rate). This resulted in a bilayer periodicity of 40 nm (in the RMS) and an overall Al content of 40 at.%. Subsequently, the samples were heat treated in a rapid thermal furnace (Jet First Joint Industrial Processor for Electronics) under argon atmosphere. For this annealing process, the heating rate was 30 K/s and the plateau time 30 s at 550 °C.

### 2.2. Analysis Methods

To investigate the microstructural features of the Al/Ni RMS on flat silicon and Al/Ni 3D-RMS, cross-section views in two directions—longitudinal and perpendicular to the multilayer planes—were prepared by focused ion beam (FIB) nanomilling (Auriga 60, Zeiss). Micrographs were obtained using the integrated scanning electron microscope (SEM) with an operating voltage of 5 kV, measurements made in tilted images were multiplied by a correction factor to obtain the correct dimensions. In addition, TEM micrographs and SAED patterns were obtained by using the TEM JEOL 2011 with an $LaB_6$ cathode and an operating voltage of 200 kV. TEM samples were prepared using the in situ lift out technique in a FIB/SEM (Helios Nanolab 600, FEI). SAED patterns were analyzed by using the software CrysTBox [21]. The analysis of the diameter of the surface structures was performed using MountainsMap®, version 7.4.9391 software with the operator motifs detection and analysis according to ISO 25178 detection algorithms (segmentation by watersheds) [22]. In order to quantify the curvature of the deposited layer due to the surface shape of the nanoneedles, the curvature of the layers was determined using the ImageJ software. Before and after RTA, both RMS and 3D-RMS, were examined using a Bruker D5000 Theta–Theta X-ray diffractometer equipped with a Cu K$\alpha$ ($\lambda$ = 0.15418 nm) radiation source used at 40 kV and 40 mA, working in Bragg–Brentano mode at speed of 1 s per step and using sampling steps of 0.02°. The diffraction patterns were analyzed in the range of 20°–100° to identify crystalline phases by using a software package (Bruker DIFFEAC.EVA V5.1) [23]. Furthermore, calorimetric investigations of the as-deposited RMS were carried with a power-compensated Perkin Elmer Differential Scanning Calorimeter 8500 using Al pans under a constant high-purity Ar flow (99.99 mol.-%)

of 20 mL min$^{-1}$. The samples were continuously heated from 300 to 823 K with a constant rate of 0.333 K s$^{-1}$. A second run with the reacted material under identical conditions was used to determine baselines that could be subtracted from the first up-scan. The calorimeter was calibrated regarding its sensitivity and temperature by measuring the melting temperatures and melting enthalpies of In and Zn. Free-standing thin films of the RMS with planar morphology were obtained by mechanically peeling the thin film from the Si substrate. Several cuts of the film were stacked over each other in order to provide a sufficient mass (1.88 mg) for a good signal-to-noise ratio. However, for the 3D-RMS, the thin film could not be detached from the substrate. In this case, the multilayer thin films were measured with the Si substrate above and stacked on top of each other to attain the best possible signal-to-noise ratio. In this case, the total mass of the deposited multilayers was estimated by measuring the surface area of the used sample pieces by light microscopy and consequently calculating the total mass by using the mass densities of Ni and Al.

## 3. Results

### 3.1. Microstructural Investigation of the Parent Layer

#### 3.1.1. Morphology on Flat Wafer

At first glance, the multilayers on the flat wafer have a smooth appearance with a mirroring surface quality. A top view of the multilayers deposited on a flat silicon wafer is depicted in Figure 2a, which reveals a rough surface of the RMS. The structure of the system in a cross-section view is shown in Figure 2b,c. The silicon substrate can be seen at the bottom of the picture and the carbon layer at the top. Carbon was used as a protection layer during sample preparation. Furthermore, the Ni and Al layers are displayed as bright and dark planes due to the Z-contrast. The increase of the multilayer roughness with increasing distance from the substrate can be observed in Figure 2b. While the first deposited layer of nickel reproduces the flat topology of the substrate as seen in Figure 2c, the Al layers seem to present interruptions and an increase in the roughness. Resulting defects can be observed at the top of the thin film in Figure 2a,d. The total thickness of the system is 4 μm.

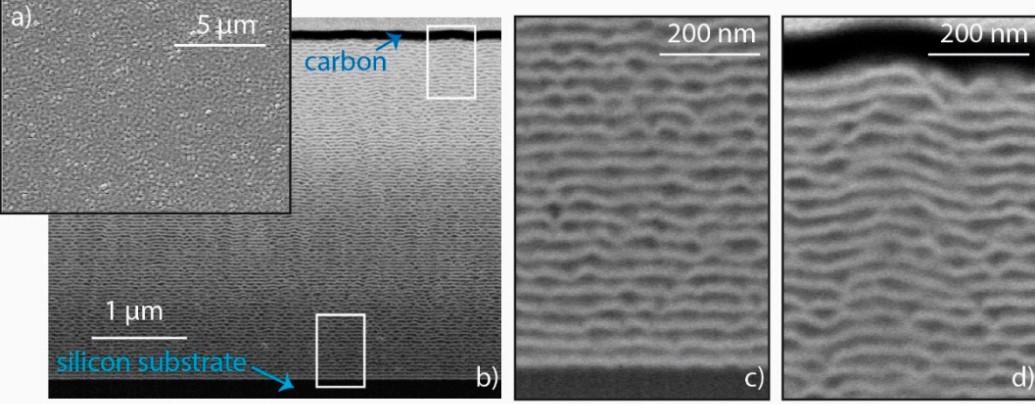

**Figure 2.** Scanning electron microscopy pictures of the planar Ni/Al multilayers: (**a**) Secondary electron image of the surface of the planar multilayer thin film from the top view. The material contrast images in (**b–d**) display the multilayer thin film in a tilted view of a FIB cross section (angle of tilt: 36°) from different positions and varying magnification.

#### 3.1.2. Morphology on Needles

Compared to the RMS on a flat silicon substrate, the RMS on Si-needles has a frosted appearance when optically investigated by eye. Figure 3a shows a top view of the 3D-RMS where domains with diameters ranging from 500 nm to 2 μm are observed. This surface differs drastically from the surface obtained on the flat silicon in Figure 2a. Considering that the deposition parameters were the same for both systems, it is possible to attribute the change in morphology of the RMS to the presence of silicon needles. The complex

topography of the substrate with needles, which vary in length and diameter, as can be seen in Figure 3b, generates heterogeneity in the speed of growth since the local peaks collect more material than the valleys, resulting in a so-called shadow instability [24]. This promotes the formation of independent conical structures that originate at the tips of silicon needles and are composed of Ni and Al layers developing 3D microstructures of Al/Ni reactive multilayers. These domains vary from one to another in size, shape, radius of curvature and, in some cases, the total number of multilayers. The domains originating on the longest silicon needles seem to grow preferentially. However, they coalesce at some points due to the formation of bridges and gaps during the growth of the film. This effect could be attributed to the shadow effect, a low diffusion coefficient, and to the flux density of the deposited material [24–26]. Despite the deposition parameters being the same for both samples, the total thickness of the multilayers deposited on flat Si was 4 µm, while the vertical length of the domains on black Si (3D-RMS) was on average 5 µm, measured from the tip of the silicon needle where the structure originates to the last layer on the surface. Due to the shadow effect, several empty spaces were created between the microstructures and this allowed the domains to reach a larger total thickness; however, the amount of material per unit area is the same for both cases. Figure 3c,c.1 shows how the domains grow laterally until they make contact with the neighboring domains, where in some cases they coalesce with each other or in others they continue to grow independently.

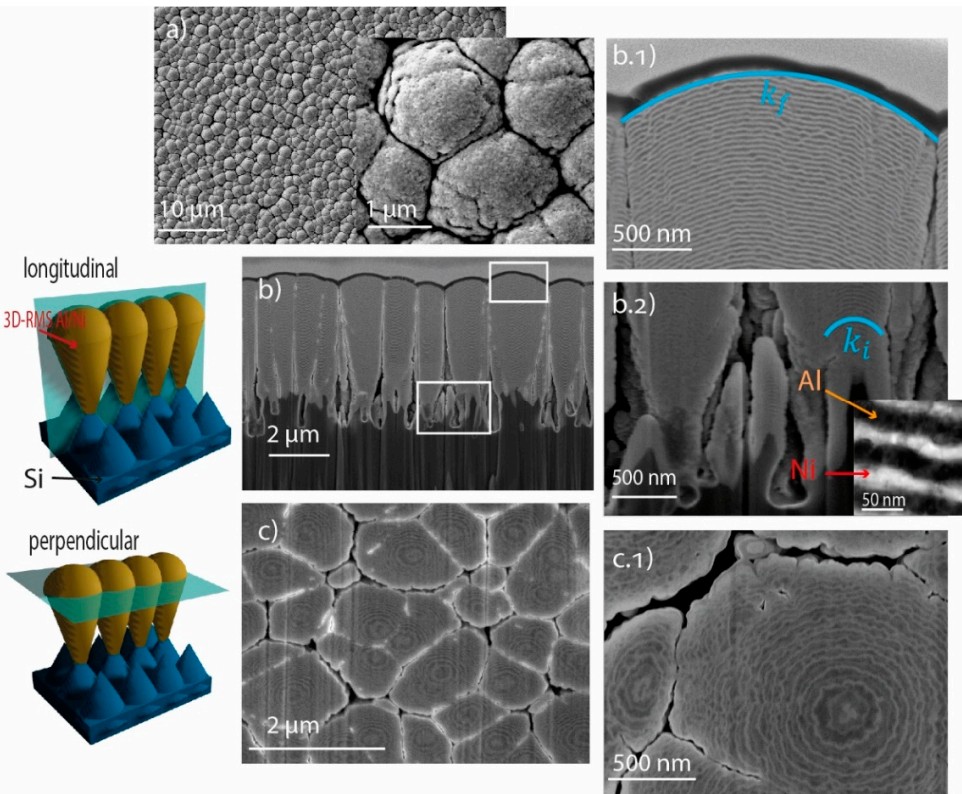

**Figure 3.** Scanning (**a,b,b.1,b.2,c,c.1**) and transmission (inset in (**b.2**)) electron microscopy images of the Ni/Al multilayers deposited on Si needles (the images were taken at the following tilt angles: (**b,b.2**) at 36° and (**c**) at 52°). The schematic representation of the mace shape on the left side illustrates the approximate region from which the samples were taken via FIB for subsequent SEM analysis of the cross-section and top-section view.

### 3.2. Size and Geometrical Analysis

The analysis of the size of the 3D-RMS displayed in Figure 4b was obtained from the micrograph in Figure 4a in an area of 2200 µm², where a total of 1200 domains were counted and the minimum and maximum diameter are 0.3 and 3.3 µm respectively. As a

result, the distribution graph of the top radius of the mace structures depicted in Figure 4b has been fitted with a log-normal distribution curve, where the calculated mean diameter is 1.4 μm with a standard deviation of 0.44 μm. The log-normal size distribution of cluster or grain sizes is frequently observed as a result of crystallite size distributions of solids; this distribution has been attributed to time-dependent growth and nucleation kinetics [27,28]. In this work, the distribution of diameter size can also be influenced by the variation in the size and diameter of the silicon needles, preferential growth, and coalescence of domains.

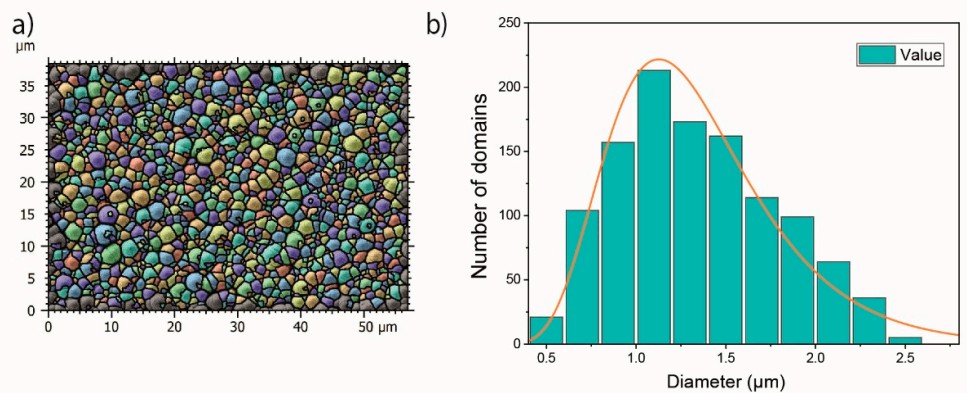

**Figure 4.** (**a**) Size of the multilayer domain structure in top-view, analyzed by a watershed filter in MountainsMap software [22], used to obtain the domain size log-normal distribution shown in (**b**).

Figure 3b.1,b.2 reveals a change of the curvature of the deposited layer due to the topography of the nanoneedles ($k_i$ and $k_f$), where $k_i$ is assigned to the smallest structures near the substrate surface and $k_f$ to those above. To quantify the curvature of the layers, the following relationship is used:

$$k = \frac{1}{R} \qquad (1)$$

where $k$ represents the curvature of the multilayer and $R$ is the radius of the arc; $k$ is calculated at different points, as shown in Figure 5a,b. This analysis was carried out for the domains that grew on the silicon needles and reached the surface without presenting interruptions during their growth. The domains that did not reach the surface or those that coalesced with each other were affected by other variables.

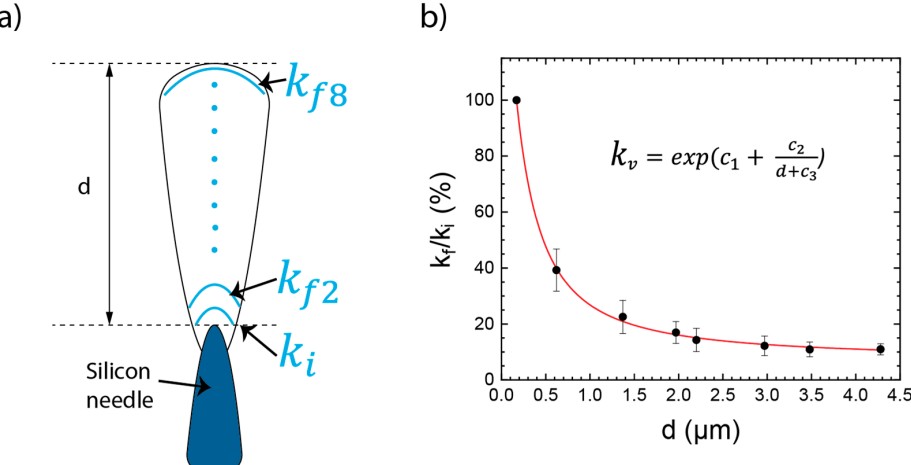

**Figure 5.** (**a**) Schematic illustration of the morphology of the 3D-RMS on top of the black Si needles displaying the different parameters taken for evaluation of the normalized curvature. (**b**) Normalized curvature as a function of distance to the first layer after evaluation of 4 different domains.

The initial curvature $k_i$ was obtained at about 170 nm from the tip of silicon needle where the growth of the structure originates (Figure 3b.2), $k_f$ was taken at different points in relation to the distance $d$ up to the last layer of the system, as seen in the Figure 5a schematic illustration. Subsequently, the normalized curvature $k_v$ was calculated using the following relation for variable distances $d$:

$$k_v = \frac{k_f}{k_i} \tag{2}$$

where $k_v$ was calculated in eight points for four domains, randomly chosen from Figure 3b, with $k_f$ at different distances $d$ from the tip of the needle. The values obtained for $k$ and $k_v$ are shown in Table 1.

**Table 1.** Curvature measurements made at various distances $d$ from the tip of the needle in different domains; $k_v$ values were obtained following Equation (2).

|  | $d$ | $k$ (1/µm) | | | | $k_v$ (%) | | | | | |
|---|---|---|---|---|---|---|---|---|---|---|---|
|  |  | 1 | 2 | 3 | 4 | 1 | 2 | 3 | 4 | Mean | SD |
| $ki$ | 0.17 | 15.55 | 5.18 | 12.19 | 9.00 | 100 | 100 | 100 | 100 | 100 | 0 |
| $kf_1$ | 0.62 | 5.09 | 1.91 | 4.56 | 4.51 | 32.5 | 36.8 | 37.4 | 50.1 | 39.3 | 7.518 |
| $kf_2$ | 1.37 | 2.57 | 1.42 | 2.24 | 2.51 | 16.6 | 27.4 | 18.4 | 27.85 | 22.5 | 5.907 |
| $kf_3$ | 1.97 | 2.06 | 1.14 | 1.77 | 1.63 | 13.3 | 21.9 | 14.5 | 18.1 | 16.9 | 3.805 |
| $kf_4$ | 2.20 | 1.80 | 1.02 | 1.29 | 1.36 | 11.6 | 19.8 | 10.6 | 15.1 | 14.3 | 4.163 |
| $kf_5$ | 2.97 | 1.41 | 0.86 | 1.17 | 1.22 | 9.1 | 16.5 | 9.6 | 13.6 | 12.2 | 3.499 |
| $kf_6$ | 3.48 | 1.33 | 0.74 | 1.09 | 1.07 | 8.6 | 14.2 | 8.97 | 11.9 | 10.9 | 2.620 |
| $kf_7$ | 4.28 | 1.27 | 0.67 | 1.01 | 0.97 | 11.9 | 12.9 | 8.2 | 10.8 | 10.9 | 2.001 |

As represented in Figure 5b, the data in Table 1 was fitted with the following exponential decay function:

$$k_v = \exp\left(c_1 + \frac{c_2}{d + c_3}\right) \tag{3}$$

where constants $c_1 = 1.89$; $c_2 = 2.38$ µm, and $c_3 = 0.71$ µm fit the equation for the resulting curve.

The growth of the nanostructures starts on the top of the needles. Owing to this, the multilayers present a curvature that reduces by 90% for the layer that is farthest from the first layer $k_i$. This trend can be seen in Figure 5b.

The graph shows that the influence of the nanostructure surface tails off exponentially with increasing thickness. Therefore, it is assumed that the strongest change in transformation behavior of the RML can be observed close to the substrate surface.

### 3.3. Characterization of the Microstructure after RTA

Both 3D-RMS on silicon needles and standard RMS on flat silicon were treated by rapid thermal annealing at 550 °C. During the process the heat radiation was used to initiate the phase transformation by solid diffusion. The multilayers deposited on flat Si after RTA peeled off from the substrate; however, the 3D-RMS remained attached to the substrate, this effect was already observed in previous investigations demonstrating the improvement in the adhesion of multilayers on silicon needles [18]. In order to verify the influence of the structure on the phase transformation, XRD analysis was performed. Figure 6 depicts the XRD patterns of both systems before and after RTA.

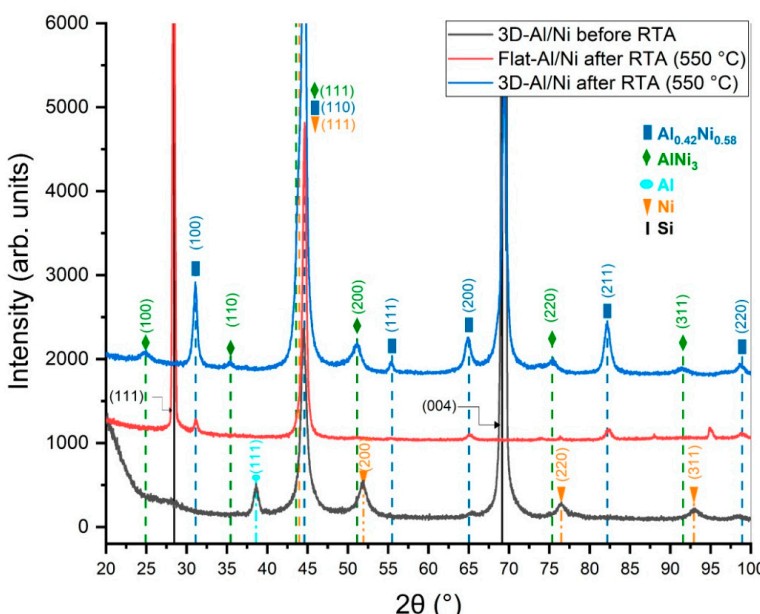

**Figure 6.** XRD diffraction patterns of 3D-RMS before and after RTA compared with standard RMS after RTA.

As deposited, in both systems the XRD patterns are similar, showing the peaks corresponding to the parent materials of Al fcc (PDF 02-004-0787) and Ni fcc (PDF 02-004-0850), where the peaks of Al (111) and Ni (200) are evident. It is also possible to observe the peaks of Si (PDF 02-027-1402), in the case of the flat Si the peak of (111) and in the case of Si needles the peak of (004). After the heat treatment, the peaks of Ni and Al disappear and the peaks corresponding to the B2 phase $Al_{0.42}Ni_{0.58}$ (PDF 02-044-1267) are present. However, a different pattern is observed in the case of the 3D-RMS after RTA, which exhibits, in addition to the peaks corresponding to B2 phase $Al_{0.42}Ni_{0.58}$, the peaks of the $AlNi_3$ phase (PDF 02-065-0144) [29]. The presence of this phase was already observed in multilayers with an excess of nickel in the atomic composition [30], and also near to the interface when the diffusion was not completed [31]. In the micrograph after RTA (Figure 7c,d), in the perpendicular section it is possible to see the traces of the multilayers, which indicates the presence of two phases (Figure 7c). However, in deeper layers, in the perpendicular section close to the Si needles, the layers are diffuse, which could indicate complete diffusion in this area. In order to reveal the phases formed in different areas of the domain, SAED analysis was carried out.

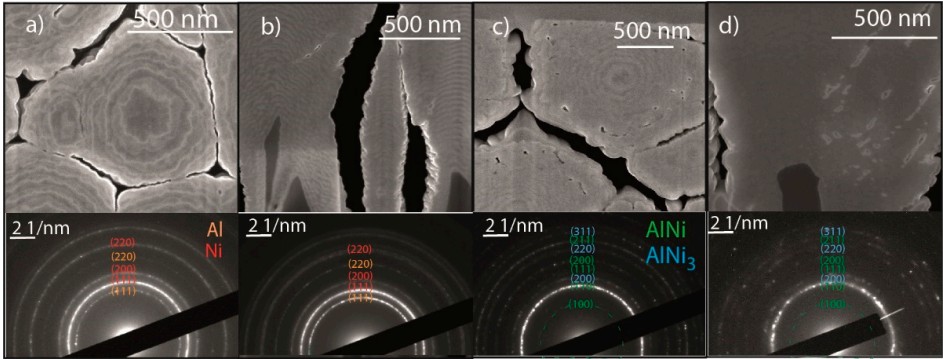

**Figure 7.** Scanning electron microscope of the 3D-RMS in a cross-section view for the as-deposited RMS (**a**,**b**) and after treatment with the RTA (**c**,**d**) up to 550 °C (the images were taken at the following tilt angles: (**a**) at 0°, (**b**–**d**) at 51° and (**c**) at 53°). The insets display the diffraction contrast images of the respective picture (**a**–**d**). The diffraction rings are indexed to certain planes and phases in every inset.

SAED analysis was performed before and after RTA and both perpendicular and longitudinal sections were analyzed. As expected before RTA, ring patterns (Figure 7a,b) were obtained, which are characteristic of polycrystalline structures. Patterns corresponding to Al fcc and Ni fcc were obtained in both areas, at a distance of 3 μm (perpendicular section) and 1 μm (longitudinal section) from the silicon needle tip. After RTA (Figure 7c,d), the Bragg reflex of Al (111) disappears in both areas and new diffraction rings appear. These new patterns correspond to AlNi (B2) and AlNi$_3$ (L1$_2$). During RTA, the Al concentration will decrease in the interdiffusion region because the diffusion of Ni in Al occurs preferentially, since the necessary activation energy for diffusion of Al in Ni is about two times the value for the diffusion of Ni in Al [32]. In both areas, closer to the needles and closer to the surface, the presence of both phases AlNi and AlN$_3$ is observed. These results confirm the presence of both phases in areas close to the silicon needles and in areas close to the surface.

### 3.4. DSC

As explained in the characterization methods, unlike the flat RMS, the 3D-RMS could not be detached from the substrate. The mass of the 3D-RMS was estimated by measuring the surface area; nevertheless, the presence of the Si substrate during the DSC analysis led to an offset of >50% in total released enthalpy compared to the planar RMS. The additional inert mass generated by the presence of the silicon nanostructured substrate leads to a deviation of the detected heat flow from the real released heat flow by the sample. Consequently, the resulting value of integrated total enthalpy release during annealing will not be correct, hence a quantitative analysis and comparison were not possible. To allow for a semiquantitatively analysis of the results based on calorimetric investigations, however, the heat flow curve of the 3D-RMS was normalized on the depth of the first exothermic peak of the corresponding heat flow curve obtained from the flat RMS (Figure 8).

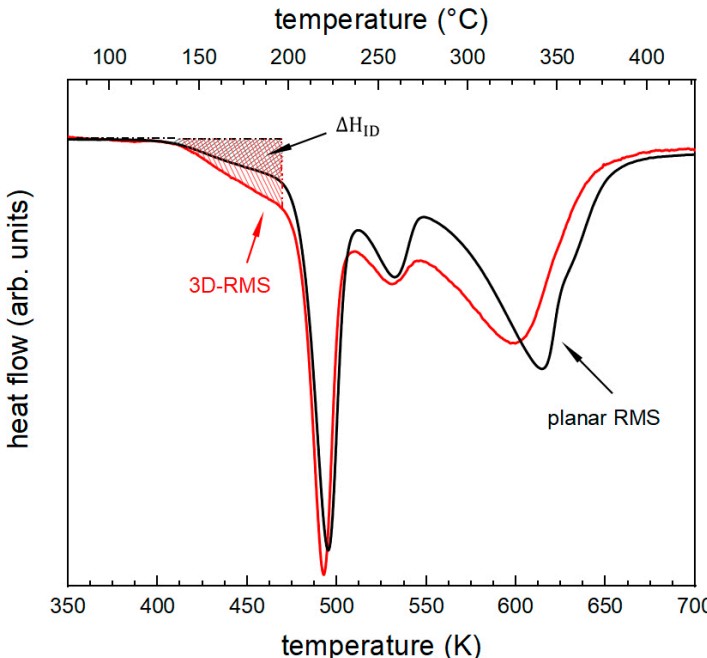

**Figure 8.** Comparison of the DSC heating scans obtained by heating with a constant rate of 0.333 K s$^{-1}$. The scans of the planar RMS and 3D-RMS are associated with the black and red curve, respectively. To compare both morphologies, the heat flow curve for the 3D-RMS was normalized on the peak depth of the first sharp exothermic event of the DSC scan with planar RMS.

A closer look at both curves reveals that for the Al/Ni 3D-RMS a higher amount of enthalpy is released during initial interdiffusion, $\Delta H_{ID}$, than for the planar RMS. Integration from the onset of interdiffusion up to 475 K, as shown exemplarily by the gray striped

area in Figure 8, yields a value of $-1639$ J/(mol of atoms) for the planar RMS. In comparison to this value, the 3D-RMS releases about 50% more enthalpy during interdiffusion. Furthermore, the peak minima of the exothermic events for the 3D-RMS are shifted to lower temperatures. This change is miniscule for the first two peaks with a difference of around 2 K. The difference for the peak temperature of the third exothermic event by 16 K is more apparent.

## 4. Discussion

### 4.1. Influence of the Black Si Surface on the Growth of the Multilayers

The morphology of the 3D-RMS differs drastically from the standard RMS (Figures 2 and 3). While continuous and compact layers grew on the flat silicon substrate, the topography of the Si needles originated the growth of conical (mace) shaped Ni/Al multilayer domains on top of the black Si. The effect of the shadow instability causes that longer needles collect more material during film growth compared to shorter needles [24]; this promotes a preferential growth of the domains that originate the needles with the highest peaks, while the domains that originate in the shorter needles stop growing when neighboring domains increase in diameter. The heterogeneity in the speed of growth generated is due to the shadow effect, which also causes formation of gaps between domains. An important effect that must also be considered is the incidence angle ($\theta$) of the arriving atoms, which will vary according the needle shape. For the atoms that reach the tip of the silicon needles, $\theta$ will be close to $0°$ while for the atoms that reach the lateral sides of the needles, $\theta$ will be close to $90°$. Previous studies demonstrated the influence of the incidence angle on the microstructure of the films deposited by magnetron sputtering [33,34]. The increase in incidence angle results not only in a higher porosity in surface morphology but also in a rougher surface [33]. Additionally, in the domains the curvature of the multilayers is observed, which is caused by the shape of the tip of the silicon needles. Figure 7d reveals the convex shape of the needle tips. During sputter deposition, a curved surface affects the coating's thickness, generating a lateral gradient with the maximum thickness at the center of the tip and the minimum thickness at the corners [34]. Due to the curvature, atoms will arrive at more grazing incidence for points farther from the center of the tip. This results in an increase of interfacial roughness of the multilayer in comparison with the multilayers deposited on flat surfaces [35]. Figure 5b depicts the multilayer's curvature changes with the growth of the domain. Therefore, it can be expected that the curvature will have a greater impact on the morphology of the first layers than in the last layers where the curvature decreases.

### 4.2. Effect of the Curvature of Al/Ni Multilayers on the Phase Transformation

In previous studies by other research groups, the influence of substrate topography on the phase transformation of reactive multilayers was demonstrated [15]. This occurs because the surface topography of the substrate affects the surface topography, the compactness, and the interphase roughness of as-deposited multilayers. These properties have a great influence in the phase transformation. For this reason, it is necessary to evaluate all the changes generated by the complex topography of the silicon needles. XRD and SAED patterns of the 3D-RMS confirmed the presence of two phases, $Al_{0.42}Ni_{0.58}$ and $AlNi_3$ after RTA, while standard RMS after heat treatment became a single phase $Al_{0.42}Ni_{0.58}$, indicating the uniform transformation of the continuous multilayer. The comparison of DSC measurements for both systems in Figure 8 discloses a sequence of exothermic events that can either correspond to the formation of a new phase or the growth of an existing phase [36]. The number of exothermic events during the phase transformation is the same for both systems. However, DSC curves reveal that for the Ni/Al 3D-RMS a higher amount of enthalpy is released during initial interdiffusion, $\Delta H_{ID}$, than for the planar RMS. This behavior can be explained considering the curvature of the 3D-RMS, which not only increases the interfacial density per unit mass [15], but also causes the increase of the interfacial roughness [35]. These characteristics of multilayers have been shown

to favor the atomic interdiffusion of Ni into the Al region [15,30]. A high diffusion flow will promote the formation of an oversaturated solid solution of nickel in aluminum at the Al/Ni interface, which is then transformed into the nonequilibrium $Al_9Ni_2$ or directly in $Al_3Ni$ phase interlayer. The metastable $Al_9Ni_2$ phase can be transformed into $Al_3Ni$ or $Al_3Ni_2$, which later, with the addition of Ni, will be transformed into B2-AlNi [30]. The crystallization of AlNi in the interface area will act as a physical diffusion barrier blocking the flow of nickel diffusion, even at higher temperatures than the melting temperature of Al [16,37]. Thus, it can be speculated that the excess of nickel will be trapped between the $Al_{0.42}Ni_{0.58}$ interlayers instead of diffusing continuously. At the same time, a small amount of aluminum can also diffuse across the interface into the Ni region and distribute heterogeneously [16], with increasing temperature and sufficient time, the high solubility of Al in Ni could promote the formation of $AlNi_3$. The formation of these two stable phases agrees with the DSC results, which show that at 700 K the reaction was complete and no remaining exothermic events were detected. However, it is difficult to predict the solid-state reaction sequence to form the final $Al_{0.42}Ni_{0.58}$ and $AlNi_3$ phases. Therefore, to reveal the sequence of reactions during the transformation of phases, it will be necessary to perform in situ XRD or quenching the phase transformation at selected intermediate temperatures.

The presence of pores in the structure, boundaries of the 3D columns, and gaps formed between the pores all act as a diffusion barrier preventing chemical and thermal diffusion during RTA. These defects generated by the shadowing hinder a uniform and complete diffusion throughout the sample, hampering the reaction of the 3D columns with each other. However, the heat provided by the radiation source placed over the sample during the RTA is uniform through the sample. It was also demonstrated with the flat Al/Ni RMS that 30 s and 500 °C give the Al/Ni system sufficient time and energy to react completely, which was also observed in previous investigations, even with a 300 °C temperature [18]. This agrees with estimations of the thermal diffusion length:

$$< x > = \sqrt{\alpha \cdot t} \tag{4}$$

where <x> is the average diffusion length, alpha is the thermal diffusivity in $m^2/s$, and *t* is the time scale considered. The metallic multilayer that conform the 3D-RMS and the doped silicon wafer are excellent heat conductors and, based on literature, alpha for the multilayers and the substrate can be approximated as $\sim 50 \times 10^{-6}$ $m^2/s$ [38,39]. Considering a timescale of 10 ms, this yields a thermal diffusion length of ~700 μm, two orders of magnitude larger than the RMS or 3D-RMS thickness. We therefore assume that the thermal gradient along the multilayer growth direction can be neglected. Consequently, it is possible to expect that each 3D Al/Ni column would react independently and generate a complete diffusion of the components through each domain, since the atomic diffusion begins at the Al/Ni interface and propagates normal to the layers [1]. XRD and SAED results show that each 3D Al/Ni column reacted, forming two stable phases with no pure metal residue observed. Additionally, Figure 7c,d shows diffuse stripes in the structure of the 3D columns after the heat treatment, which confirms the presence of two phases in the sectional views of the domains, despite the flat RMS that reacted and formed a single phase. Therefore, it is possible to attribute the formation of the two phases to the deformation of the layers, the columnar growth, crystallographic defects, and probably local compositional variations produced by morphology of the Si needles together with the shadowing effect during the sputtering deposition. Further studies are necessary to elucidate the impact of each of these factors on the phase transformation.

## 5. Conclusions

3D-RMS were characterized by scanning and transmission electron microscopy before and after RTA and compared with standard RMS. The difference in the morphology of these two systems was attributed to the shadow effect generated by the topography of nanostructured black silicon. XRD and SAED patterns revealed that the novel morphology of 3D-RMS during RTA promotes the formation of the two stable phases $Al_{0.42}Ni_{0.58}$ and

AlNi$_3$ for a system with 60% Ni atomic composition. Qualitative differential scanning calorimetry analysis of the 3D-RMS indicates that the curvature of the nanoscale multilayer favors the atomic diffusion in the early stage of the phase transformation. This study provides useful information for better understanding the microstructure and morphology of the 3D-RMS and its influence on the phase transformation during RTA and DSC. Further studies are necessary to quantify the energy release of the 3D-RMS and to identify the solid-state reaction sequence for both systems.

**Author Contributions:** Conceptualization, Y.H.S.C., P.S. and H.B.; methodology, Y.H.S.C. and H.B.; investigation, Y.H.S.C., S.S.R., C.S., K.J. and J.S.; supervision, P.S., F.M. and I.G.; writing, Y.H.S.C. and S.S.R.; review and editing, H.B., C.P. and P.S. All authors have read and agreed to the published version of the manuscript.

**Funding:** This study was supported by the Deutsche Forschungsgemeinschaft (DFG grants Scha 32/30 and Scha 632/29), I.G. and S.S.R. acknowledge financial support from the German Research Foundation (DFG) through grant GA 1721/3-1. C.S. and F.M. acknowledge funding from DFG grant MU 959/48-1. Support by the Center of Micro- and Nanotechnologies (ZMN), a DFG-funded core facility of the TU Ilmenau, is gratefully acknowledged.

**Institutional Review Board Statement:** Not applicable.

**Informed Consent Statement:** Not applicable.

**Data Availability Statement:** The data that support the findings of this study are available from the corresponding author, Y.H.S.C., upon reasonable request.

**Acknowledgments:** The authors express deep thankfulness to all the staff of the Center of Micro- and Nanotechnologies (ZMN), especially to Joachim Döll, Henry Romanus, and Manuel Oliva, for their support with some experiments (sputtering and characterization analysis).

**Conflicts of Interest:** The authors declare no conflict of interest.

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
