# Peer review of "Phase Transformation and Characterization of 3D Reactive Microstructures in Nanoscale Al/Ni Multilayers"

_applsci, doi:10.3390/app11199304_

Round 1

Reviewer 1 Report

The paper is nicely written and presents a significant amount of work on the subject that certainly is of interest for the readership of the Applied Sciences. I'm not a specialist in Ni-Al alloys but it seems that the work presented here is novel and original.

The authors put quite an effort to present the nice and thorough analysis of the curvature of deposited nanolayers as a function of the height of each microcolumn. I'm not however convinced by the results presented here that it is necessarily a curvature of the deposited films that leads to the differences between the films on flat silicon and on 3D structured surface. It seems to me that other factors like the shape of the 3D columns, the fact that they have very little contact with substrate and that they are isolated from each other should also play the important role in the heat dynamics and as a result in the crystalline structure of AlNi alloy. I don’t see in the manuscript the reasoning why these parameters were eliminated, and the curvature was given the main role in this discussion.  There is no comparison between 3D samples but with different curvature, for example. It worth to note also that at nanoscale the layers deposited on flat Si are quite curved, especially when they are far from the surface. I would ask authors to put some discussion on this subject to this paper and probably modify the conclusions in this sense.

Other small remarks:

Fig. 4b: The y-axis title should be kf/ki, not vice versa

Fig. 5: It would be proper to add the symbols (circle, diamond...) used for the identification of different peaks to the legend to help those who will print the paper.

Lines 270, 272, 383 and probably more: Most of the time the predominant phase is marked as Al_0.42_Ni_0.58, but in few places it is Al_0.4_Ni_0.58. It would be good to keep a consistent notation.

Author Response

Dear Reviewer,

thanks for the feedback and for the time invested in this manuscript, the suggestions have been taken into account and they helped us to improve the manuscript.

Please see the attachment to find our responses

Reviewer 2 Report

The manuscript reports about 3D reactive microstructures obtained with nanoscale  Al/Ni multilayer systems grown by magnetron sputtering, as grown and submitted to rapid thermal annealing. The properties of 3D structures were compared with the ones of reactive multilayers systems grown (at same sputtering conditions) on flat Si substrate. The study focuses on the structural, morphological and phase transformation analysis of the two microstructures which differ due to the different substrates’ topography. The manuscript is interesting, well written and the work described with scientific rigour. A few minor comments are reported below.

  • The morphology of the nanostructured Si substrate could have been shown while describing the Si needle fabrication. In fact, the substrate morphology is inferred only later from Figure 2 b.
  • In the Materials and Methods Section (page 2 line 96), please describe how you controlled the Ni and Al single layer thickness used to fabricate the multilayer stack.
  • Please specify the ambient where the rapid thermal annealing was carried out.
  • Please check if at page 6 line 222 the referred figure is Figure 3b instead of Figure 2b.
  • Description of DSC measurement (page lines 301-312) could be improved to make it more clear.
  • Please correct Figure 1.2 into Figures 1 and 2 (page 10 line 333).

Author Response

Dear Reviewer,

thanks for the feedback and for the time invested in this manuscript, the suggestions have been taken into account and they helped to improve the manuscript.

Please see the attachment to find our responses
